# Gadolinium-Doped Carbon Nanoparticles with Red Fluorescence and Enhanced Proton Relaxivity as Bimodal Nanoprobes for Bioimaging Applications

Dariia U. Musaeva [1], Alexey N. Kopylov [1], Alexander V. Syuy [2], Valentyn S. Volkov [2], Nikita D. Mitiushev [3], Olga S. Pavlova [4,5], Yury A. Pirogov [5], Andrey N. Baranov [6] and Victor Yu. Timoshenko [5,*]

1   Phys-Bio Institute, National Research Nuclear University MEPhI, 115409 Moscow, Russia; d.ariya@bk.ru (D.U.M.); lex.kopylov@gmail.com (A.N.K.)
2   Center for Photonics and 2D Materials, Moscow Institute of Physics and Technology, 141701 Dolgoprudny, Russia; alsyuy271@gmail.com (A.V.S.); vsv.mipt@gmail.com (V.S.V.)
3   Faculty of Materials Science, Lomonosov Moscow State University, 119991 Moscow, Russia; nikita.mit55@gmail.com
4   Faculty of Fundamental Medicine, Lomonosov Moscow State University, Leninskie Gory 1, 119991 Moscow, Russia; ofleurp@mail.ru
5   Faculty of Physics, Lomonosov Moscow State University, Leninskie Gory 1, 119991 Moscow, Russia; yupi937@gmail.com
6   Faculty of Chemistry, Lomonosov Moscow State University, Leninskie Gory 1, 119991 Moscow, Russia; anb@inorg.chem.msu.ru
*   Correspondence: timoshen@physics.msu.ru

**Abstract:** Carbon-based nanoparticles (CNPs) have demonstrated great potential in biomedical applications because of their unique physical and chemical properties, and excellent biocompatibility. Herein, we have studied two types of CNPs with gadolinium (Gd) impurities (Gd-CNPs), which were prepared by microwave synthesis (MWS) and hydrothermal synthesis (HTS), for potential applications as photoluminescent (PL) labels and contrast agents in magnetic resonance imaging (MRI). The prepared Gd-CNPs were investigated by means of transmission electron microscopy (TEM), Fourier-transform infrared spectroscopy, UV–visible absorption spectroscopy, and magnetic-resonance relaxometry, which allowed us to reveal specific features and functional properties of the prepared samples. While the TEM data showed similar size distributions of both types of Gd-CNPs with mean sizes of 4–5 nm, the optical absorption spectroscopy showed higher absorption in the visible spectral region and stronger PL in the red and near-infrared (NIR) spectral regions for the MWS samples in comparison with those prepared by HTS. Under green light excitation the former samples exhibited the bright red-NIR PL with quantum efficiency of the order of 10%. The proton relaxometry measurements demonstrated that the HTS samples possessed longitudinal and transverse relaxivities of about 42 and 70 mM$^{-1}$s$^{-1}$, whereas the corresponding values for the MWS samples were about 8 and 13 mM$^{-1}$s$^{-1}$, respectively. The obtained results can be useful for the selection of appropriate synthesis conditions for carbon-based nanoparticles for bimodal bioimaging applications.

**Keywords:** carbon; nanoparticles; dots; gadolinium; photoluminescence; contrast agent; MRI; relaxivity; bioimaging

## 1. Introduction

Since their discovery in 2004 [1], small carbon-based nanoparticles (NPs) or as-called carbon dots (CDs) have attracted significant interest because of their bright fluorescence [2,3] and possible applications in photonics [2–7], sensing [7–12], photocatalysis [13–16], and biomedical treatments [7,17]. While CDs are usually associated with very small NPs with sizes from 1 to 10 nm [1–10], the mean sizes of larger carbon NPs (CNPs) can be above 10 nm [10–12]. While such carbon-based NPs can be prepared by using different bottom-up and top-down

approaches [1,6,7], the bottom-up method of the synthesis via thermal decomposition of carbon-containing precursors in microwave [1,9] and hydrothermal [10–12] reactors are most convenient, flexible, and scalable.

Both CDs and CNPs exhibit many advantageous properties such as high photoluminescence (PL) quantum yield, photostability, and spectral tunability [5–7,10]. These features, together with low toxicity, make CDs and CNPs highly attractive for both the bioimaging application [4,6,7,10] and therapies, e.g., photo- and radiofrequency hyperthermia [7,17].

Among many potential bioimaging applications of CNPs, their use as multimodal fluorescence imaging probes and simultaneously as contrast agents in magnetic resonance imaging (MRI) has gained considerable attention [18–24]. It was found that gadolinium-doped CDs (Gd-CDs) with efficient PL could be simultaneously used as contrast agents (CAs) for MRI [18,19]. Dual modality imaging by using Gd-CDs with efficient PL and MRI contrasting was demonstrated in vitro and in vivo [20–22]. Gadolinium and bismuth co-doped CDs were explored for multimodal imaging in fluorescence, computer X-ray tomography, and MRI modes [23]. Moreover, Gd-CDs were demonstrated to be a promising theranostic agent for fluorescence and MRI-guided cancer phototherapy [24].

While facile syntheses of fluorescent Gd-doped CDs and CNPs by both MWS and HTS are successfully demonstrated, each technique will offer distinct advantages and challenges for further applications in optical bioimaging and MRI. Here, we present a comparative analysis of Gd-CNPs prepared by MWS and HTS methods to evaluate their performance as red fluorescent labels and efficient CAs in MRI. By elucidating the similarities and differences between the two synthesis methods, this study aims to provide valuable insights into the optimal fabrication approach for Gd-doped CDs and CNPs with enhanced imaging capabilities.

## 2. Materials and Methods

Microwave synthesis (MWS) was carried out by using a mixture of urea ($(NH_2)_2CO$) 300 mg, citric acid ($HOOC-CH_2-C(OH)COOH-CH_2COOH$) 150 mg, and gadolinium chloride ($GdCl_3 \cdot 6H_2O$) 30 mg as precursors, which were dissolved in 5 mL of deionized water (18 MΩ·cm). Then, synthesis was carried out in a microwave reactor (Anton Paar Monowave 450, Strašnice, Czech Republic) at a temperature of 180 °C with stirring at 1200 rpm for 15 min. The precursors' ratios and synthesis conditions were chosen to obtain a high yield of small CNPs with efficient red PL [18,19].

The MWS process resulted in the formation of yellow-colored solutions, which later precipitated as a white solid. Centrifugation was performed to separate the precipitate, and then 20 mL of isopropyl alcohol was added to the 5 mL solution in order to separate small CNPs and byproducts as larger amorphous graphite NPs because of a lower density and surface tension of the solvent. The supernatant of Gd-CNPs in isopropyl alcohol was selected and the solution was subsequently evaporated. Then, the dried Gd-CNPs were transferred back into water, resulting in a transparent solution with red color.

For hydrothermal synthesis (HTS) we used aqueous solutions containing urea 50 mg/mL, citric acid 25 mg/mL, and gadolinium chloride 2 mg/mL as precursors. The synthesis was carried out in a 45 mL stainless steel hydrothermal reactor with a Teflon liner at 150 °C for 3 h followed by cooling down in a natural way for 2 h. The HTS process yielded transparent suspensions Gd-doped CNPs with red color.

Prior to the structural studies and optical and MRI experiments, the prepared aqueous suspensions of Gd-CNPs were purified by dialysis in deionized water through MWCO 500 Da membranes for 6 h.

To estimate the total mass concentration of Gd-CNPs in the solution, we used a GR-202 analytical scale (A&D Company, Ltd., Tokyo, Japan) with a measurement accuracy of 0.01 mg. Then, 100–500 mL of the aqueous solution of Gd-CNPs were taken and dried on a glass substate at 80 °C for 20 min.

The Gd concentration in Gd-CNPs was estimated by means of the X-ray fluorescence analysis (XFA) with a Radian-02 X-ray diffractometer with a Cu-Kα radiation source for

the samples dried on a molybdenum substrate at 80 °C for 20 min. A calibration of the XFA method was carried out by using dried solutions of $GdCl_3$ with known Gd concentration as the reference. The measurements were carried out at room temperature in the air.

Sizes, structures, and compositions of the prepared NPs were studied by using a JEOL JEM-2100 transmission electron microscope (TEM) (JEOL, Inc., Peabody, MA, USA), which could also operate in a high-resolution mode (HR-TEM). Size distributions of Gd-CNPs were obtained from their TEM images by using free ImageJ software (version 1.53t). The elemental compositions of the prepared NPs were determined by means of the energy dispersive X-ray spectroscopy (EDX) with the same TEM apparatus equipped with an Aztech X-Max 100 EDX device (Oxford Instruments, Abingdon, UK).

To analyze the chemical bonds in Gd-CNPs, measurements by means of the Fourier-transform infrared spectroscopy (FTIR) were carried out with a Bruker IFS-66v/s spectrometer (Bruker, Billerica, MA, USA) in the range of 500–4000 $cm^{-1}$ with steps of 4 $cm^{-1}$ in the geometry of attenuated total reflection (ATR) with a diamond ATR-crystal.

Optical absorbance spectra of aqueous suspensions of Gd-CNPs in quartz cuvettes were measured by using a UV-VIS 752P spectrophotometer in the range of 250–900 nm with a resolution of 1 nm. The concentrations of the studied solutions were typically 0.1 mg/mL. PL spectra of the prepared aqueous suspensions of Cd-CNPs were measured in the spectral range from 350 to 900 nm by using a Mightex compact spectrometer (Mightex, Toronto, ON, Canada) with a CCD array. PL excitation was carried out by using either a UV LED with a wavelength of 365 nm, power 0.5 W, and beam diameter of about 10 mm or a semiconductor laser with a wavelength of 532 nm, maximal power of 0.2 W, and beam diameter of 2 mm. The PL quantum yield (QY) of prepared Cd-CNPs was estimated by using a 50 mm integrating sphere (Thorlabs Inc., Newton, NJ, USA) and a reference aqueous solution of Rhodamine 6G.

Measurements of the proton relaxation times in aqueous suspensions of Cd-CNPs were carried out using a 7 T Bruker BioSpec 70/30 USR MRI scanner. The suspensions were diluted in deionized water at various Gd concentrations and placed in 1.5 mL Eppendorf tubes. The sample temperature during the measurements was kept at 19.5 °C.

## 3. Results and Discussion

Figure 1A,B show typical TEM images of dried diluted suspensions of the Cd-CNPs prepared by MWS and THS, respectively. The MWS sample consists of nearly spherical NPs with size distribution, which can be fitted by a lognormal function with a mean size of 5 nm (right inset in Figure 1A). A HR-TEM image of a selected CNP reveals atomic planes with a lattice spacing of 0.22 ± 0.01 nm (left inset in Figure 1A). This lattice parameter is close to the interatomic distance for the $(01\bar{1}0)$ crystallographic plane of graphite [25] and it is corresponding to the in-plane periodicity in graphene-like nanosheets [26]. The same lattice spacing was observed early for CDs and CNPs synthesized by MCW [16,18], HTS [20] or similar methods at about 180 °C and slightly higher temperatures [26]. It is interesting to note that the graphene-like interlayer distances have been preserved in Gd-doped CNPs similar to the results for catalytically active CD-based nanocomposites, which were highly doped with different impurities such as aluminum and tellurium [26,27]. It means that the incorporation of both light and heavy elements does not necessarily destroy the atomic order in CDs and CNPs.

An analysis of the TEM data of HTS samples (Figure 1B) demonstrates NPs with size distribution covered by a lognormal function centered at 4 nm (right inset in Figure 1B). In contrast to the MWS samples, the HTS ones do not exhibit a prominent lattice structure as it is checked by the HR-TEM analysis (left inset in Figure 1B). This fact can be explained by the relatively low temperature of 150 °C during the HTS treatment which is not sufficient to form the crystalline order in CNP. Note that the similar HTS procedure at 200 °C resulted in the formation of small graphite NPs, which were identified by the characteristic interatomic spacing of about 0.334 nm [19]. The graphite-like CNPs seem to be not optimal

for bioimaging applications because of the nonradiative losses of light in the electrically conductive cores.

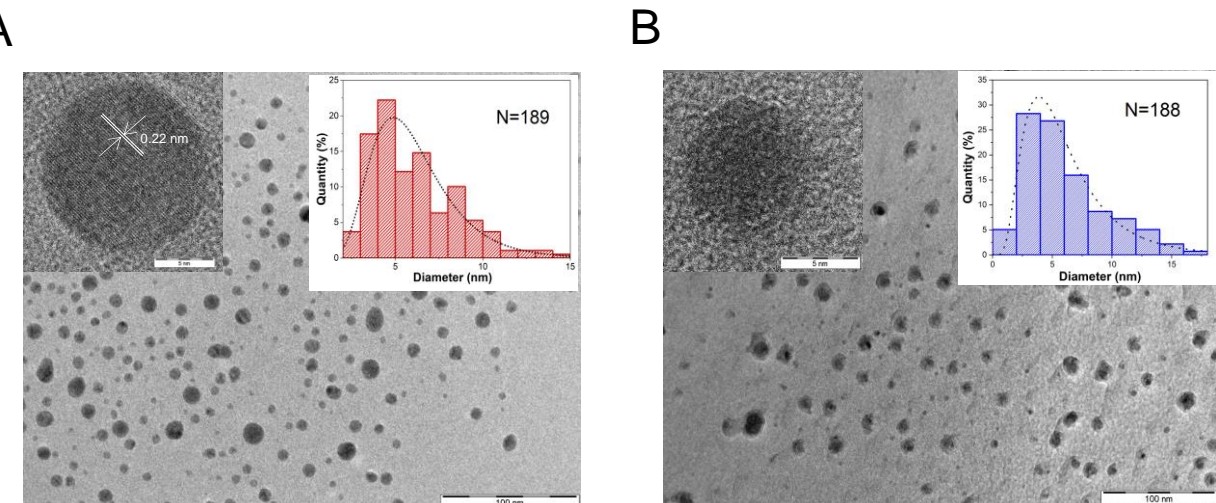

**Figure 1.** (**A**) TEM image of MWS Gd-CNPs; the left-hand and right-hand insets show an HR-TEM image of an individual NP and size distribution obtained from the TEM image analysis, respectively. (**B**) TEM image of HTS Gd-CNPs; the left-hand and right-hand insets show an HR-TEM image of an individual NP and size distribution obtained from the TEM image analysis, respectively. The size distributions were obtained by counting N = 189 and 188 particles for the TEM images of MWS and HTS samples, respectively.

According to the EDX analysis (see Figures S2 and S4 and Tables S1 and S2 in the Supplementary Materials) the Cd-content was about 0.04 and 0.5 at. % for the samples prepared by MWS and HTS methods, respectively. The higher Gd-doping level of the latter was achieved despite the lower Gd-content in the precursor solution in comparison with the MCW samples. This fact indicates the better incorporation of Gd-ions in amorphous CNPs than in the nanocrystalline ones. Note, the relatively high concentration of oxygen (14 at. %) and nitrogen (13 at. %) correlate with the higher Gd content in the HTS sample. In contrast, the MCW sample exhibits a rather low concentration of oxygen (7.8 at. %) and nitrogen (0.16 at. %) atoms. This fact can be interpreted as a predominant surface location of these atoms and Gd-impurities in the MWS Gd-CDs, while the HTS-prepared mostly amorphous samples can contain the impurities in the whole NPs' volume.

According to the FTIR spectroscopy data (Figure 2A), the chemical bonds in the prepared Gd-CNPs are mostly related to the well-known vibration modes of the carbonyl, carboxyl, and other functional groups, which contain carbon, oxygen, hydrogen, and nitrogen (Table 1). In particular, the spectrum of the HTS-prepared sample exhibits a significantly higher signal at 1557 cm$^{-1}$, which is assigned to the vibration and deformation bands of N–H groups [28]. Additionally, there is a strong signal at 1391 cm$^{-1}$ associated with the C=C bond vibrations [28]. This line is also close to the position of C-O bond vibrations (1388 cm$^{-1}$) detected early in Gd-CDs [19]. In contrast, the spectrum of the MWS sample exhibits a strong asymmetric peak at 1438 cm$^{-1}$, which can be related to the C-N bonds from aromatic amine groups and C-C vibrations of the carbon mesh frame [8,10]. Both samples exhibit similar absorption peaks in the region of 2500–3500 cm$^{-1}$, which corresponds to the superposition of the stretching vibrations of N-H, O-H, and C-H groups [29,30]. Note, the total absorption by the chemical bonds in the MWS sample is lower than that for the HTS one (see Figure 2A). For example, the absorption at 1078 cm$^{-1}$, which corresponds to the C–O and C–N groups, is five times stronger for the HTS Gd-CNPs than that for the MWS ones. This fact correlates with the higher contents of oxygen and nitrogen in the formers (see Tables S1 and S2 in the Supplementary Materials).

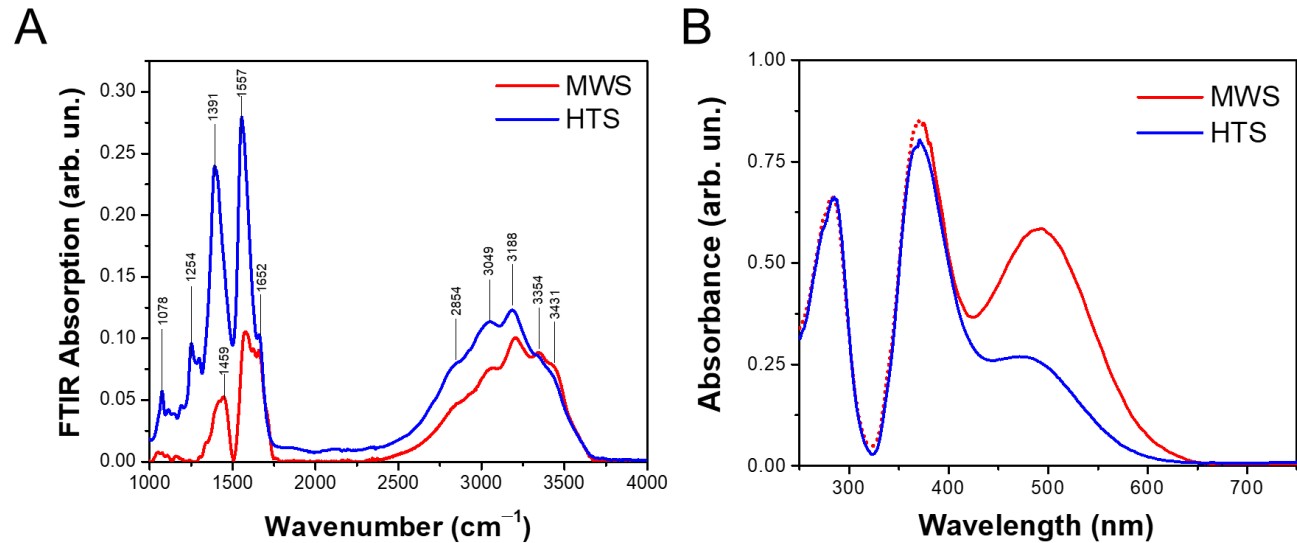

**Figure 2.** (**A**) FTIR absorption spectra of dried Gd-CNPs prepared by MWS (red curve) and HTS (blue curve); (**B**) UV–visible absorption spectra of aqueous suspensions of Gd-CNPs prepared by MWS (red curve) and HTS (blue curve).

**Table 1.** Assignment of the infrared absorption bands for Gd-CNPs.

| Wavenumber (cm$^{-1}$) | Functional Groups and Bands | References |
|---|---|---|
| 3375–3468 | N–H | [8,20,30] |
| 3060–3431 | O–H | [8,20,22] |
| 3050 | N–H | [30] |
| 2854 | C–H | [22] |
| 1652 | C=C isolated and conjugated | [29] |
| 1651 | C=O stretching | [28] |
| 1557 | N–H deformation | [29] |
| 1438 | C–N in aromatic amine and C–C of the carbon mesh frame | [8,10] |
| 1391 | C=C stretching and bending | [28] |
| 1388 | C–O | [19] |
| 1254 | C–O–C | [18] |
| 1078–1122 | C–O, C–N | [9,28,29] |

Figure 2B shows UV–visible–NIR absorption spectra of the aqueous suspensions of both types of Gd-CNPs of the same total concentration. The spectra consist of three main bands at about 280, 370, and 490 nm. While the UV absorption bands are similar for both types of Gd-CNPs, the visible absorption band is significantly stronger for the MWS sample than for the HTS one. Because the former exhibits the higher absorption via the C-N bonds from aromatic amine groups it can be supposed that the visible absorption band also originates from those amine groups [8,29].

Figure 3A shows PL spectra of the prepared aqueous suspensions of Gd-CNPs under UV excitation. The samples of both types are fluorescent mainly in the blue-green spectral range with tails in the red-NIR region. The PL intensity of the MWS sample is slightly larger than that of the HTS one. Since the UV absorption spectra of both samples are almost the same value (see Figure 2B), the difference in the PL intensity indicates the different ratio between the radiative and non-radiative processes in the MWS and HTS samples. The non-radiative losses in the latter can be related to the larger Gd-content since Gd-ions can act as quenchers of the fluorescence of CNPs as it was found for different metal cations [7,29,30]. The stronger PL QY of the MWS sample can be related to the better passivation of the non-radiative recombination sites due to the higher temperature of the synthesis. This fact correlates with the better crystallinity of the MWS Gd-CNPs (see Figure 1A), which represent mostly nanocrystalline CDs covered by thin shells of the functional groups listed

in Table 1. Note, the PL QY of the latter under the UV excitation is estimated to be about $80 \pm 10\%$ which indicates the dominant role of the radiative recombination in the MWS Gd-NPs.

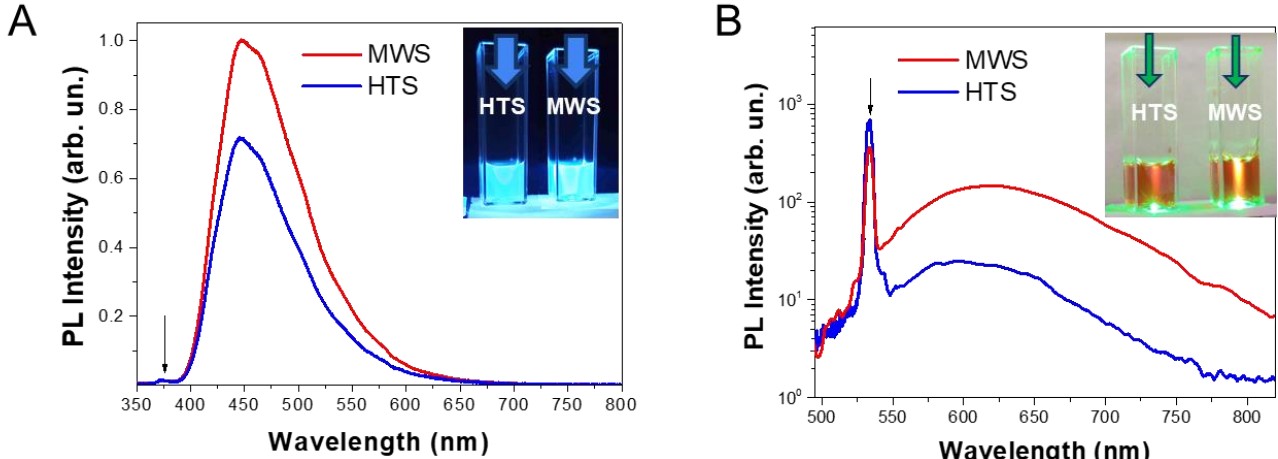

**Figure 3.** (**A**) PL spectra of the MWS (red curve) and HTS (blue curve) Gd-CNPs under UV excitation (365 nm); the excitation line is indicated by black arrow. (**B**) PL spectra of the MWS (red curve) and HTS (blue curve) Gd-CNPs under excitation at 532 nm (indicated by black arrow). Insets show photographic images of quvetes with the MWS (right hand) and HTS (left hand) Gd-CNPs excited by the UV light and laser radiation at 532 nm, which are indicated by blue and green arrows, respectively.

Being excited with the green light at 532 nm, the MWS-prepared Gd-CNPs exhibit significantly stronger PL emission compared to the HTS samples, as shown in Figure 3B. While the maxima of the PL spectra of both samples are located near 600 nm, the PL intensity of the MWS sample is about five–six times higher in the red-NIR spectral regions. The stronger PL of the MWS Gd-CNPs correlates with the stronger absorption in the visible spectral region (see Figure 2B). This fact can be explained by the higher density of the optically active states in the MWS Gd-CNPs. As for the PL QY of the latter under the used green laser excitation, it is estimated to be about $10 \pm 5\%$. Note, the observed red-NIR PL spectra of the prepared samples lie in the first optical-transparency window of biotissue [31] that is favorable for fluorescent bioimaging application of Gd-CNPs.

It is interesting to note that the PL spectra of both types of Gd-CDs under green light excitation consist of the emission with photon energy above the excitation one. Because the PL excitation was carried out with the continuous wave (CW) light of relatively low intensity, one can rule out the two-photon absorption and other non-linear optical up-conversion processes. The anti-Stokes PL under low intensive CW excitation is well known for semiconductor nanocrystals, and it is explained by phonon-assisted absorption of exciting light [32]. The up-conversion PL excitation in Gd-CDs can be related to the absorption of vibrations of the carbon mesh, carbon–oxygen, and other functional groups that required further detailed studies.

Figure 4 shows dependencies of the reciprocal values of the longitudinal ($T_1$) and transverse ($T_2$) proton relaxation times on Gd-concentration in aqueous suspensions of Gd-CNPs. The relaxation efficiencies, i.e., the longitudinal ($r_1$) and transverse ($r_2$) relaxivities, can be obtained from linear fits of the corresponding dependences on Cd concentration [19–23]. Considering the different initial Cd content in the samples prepared by MWS and HTS methods, the relaxivity dependences were obtained in the different titration degrees. It was found that the HTS-prepared sample exhibited larger values $r_1$ and $r_2$, which account for 42 and 70 mM$^{-1}$s$^{-1}$, whereas the corresponding relativities for the MWS samples were about 8 and 13 mM$^{-1}$s$^{-1}$, respectively. These differences may be attributed to the synthesis conditions, which are more temperature-controllable for the HTS process and promote desired coordination of Gd-ions on the surfaces of mostly amorphous CNPs to ensure more efficient

involvement of the formers in the shortening of the proton relaxation times. It means that the Gd-ions in the HTS samples with mostly amorphous CNPs are about five-times more efficient in the proton relaxation shortening than the same number of GD-ions in the MWS samples.

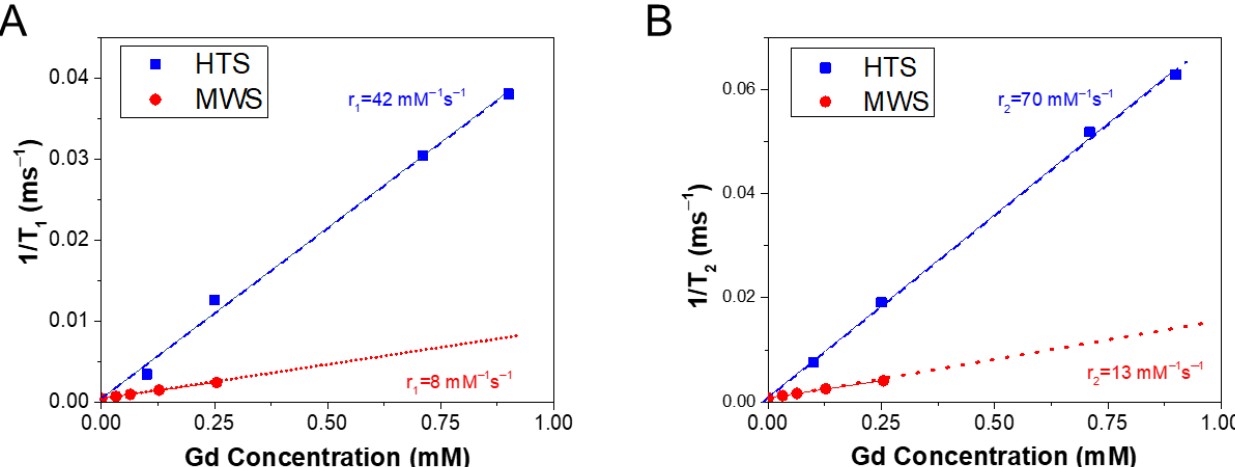

**Figure 4.** (**A**) Dependences of the reciprocal time of the longitudinal proton relaxation in aqueous suspensions of Gd-CNPs prepared by MWS (red circles) and HTS (blue squares) on Gd-concentration. (**B**) Concentration dependencies of the reciprocal time of the transverse proton relaxation for the same samples. The indicated longitudinal relaxivity, $r_1$, and transverse one, $r_2$, are obtained from linear fits of the corresponding concentration dependencies. The different titrations in the concentration dependencies for the MCW and HTS samples are related to their different initial Gd-content.

It should be noticed that the longitudinal relaxivity of the HTS-prepared Gd-CNPs is several times higher than other results on similar Gd-doped CNPs [18–21]. However, the observed relaxivity of our Gd-CNPs is four times smaller than that for 3 nm sized amorphous Cd-CDs prepared by HTS at 180 °C for 8 h, when the commercial CA, i.e., Gadovist®, was used as a precursor [22]. However, the corresponding Gd-CDs exhibited an exciting wavelength-independent green-yellow PL band centered at 500 nm [22], which did not correspond to the biotissue transparency window. The longitudinal relaxivities of our Gd-CNPs obtained by both methods are higher than those values for the commercially available Gd-based molecular CAs [33,34] and theranostic Gd-based NPs [35,36]. The high values of both longitudinal and transverse relaxivities of the prepared Gd-CNPs are probably determined by the employed synthesis and purification procedures, which ensure the desired surface arrangement and molecular coordination of $Gd^{3+}$ ins in CNPs.

### 4. Conclusions

Finally, the comparative study of Gd-doped carbon NPs synthesized through microwave and hydrothermal methods reveals their specific structural and physical properties for potential applications in bimodal fluorescence/MRI bioimaging. The results of the structure analysis and optical spectroscopy show that the MWS samples are more crystalline ordered and possess stronger absorption in the visible spectral range which results in the high photoluminescence efficiency of the red-NIR under green light excitation. This fact suggests that microwave synthesis could be preferable when targeting luminescence-based imaging techniques, enabling sensitive cellular or subcellular visualization. Based on the relaxometry measurements, the HTS sample exhibited superior MRI relaxivity properties. The relatively long duration and spatially homogeneous temperature profile during the hydrothermal synthesis allowed us to realize the efficient incorporation of Gd-ions on the surface of CNPs, leading to enhanced MRI contrast. Thus, the hydrothermal method could be advantageous for producing Gd-doped carbon NPs with excellent MRI contrasting, enabling accurate anatomical imaging in a clinical setting. The longitudinal and transverse relaxivities for the HTS Gd-CNPs are significantly larger than that for commercially



available Gd-based CAs. The divergent properties observed between the two synthesis methods highlight the importance of selecting an appropriate synthesis approach based on the desired imaging modality. If MRI contrast enhancement is the primary goal, the hydrothermal method appears to be more suitable. Conversely, for biomedical applications where luminescence bioimaging is of greater importance, e.g., at the cell level and intraoperative visualization, the microwave method may be preferable. Future research should focus on optimizing the synthesis conditions of both methods to further improve the desired imaging properties. Additionally, exploring hybrid approaches that combine the strengths of both methods could potentially yield optimized Gd-doped CNPs and CDs with enhanced bimodal imaging capabilities.

**Supplementary Materials:** The following supporting information can be downloaded at: https://www.mdpi.com/article/10.3390/app13169322/s1, Figure S1: TEM images and EDX elemental maps for HTS Gd-CDs; Figure S2: EDX spectrum of HTS Gd-CDs; Figure S3: EDS-TEM images and composition of an individual Gd-CD; Figure S4: EDX-TEM images and composition of MWS Gd-CD; Table S1: Elemental content of HTS Gd-CNPs; Table S2: Elemental content of MWS Gd-CNPs.

**Author Contributions:** Investigation and writing—original draft preparation, D.U.M.; methodology and investigation, A.N.K.; visualization and formal analysis, A.V.S.; resources, methodology, and investigation, V.S.V.; investigation, N.D.M.; investigation, O.S.P.; methodology and data curation, Y.A.P.; methodology and writing—review and editing, A.N.B.; supervision and writing—review and editing, V.Y.T. All authors have read and agreed to the published version of the manuscript.

**Funding:** A.V.S. and V.S.V. acknowledge the financial support from the Ministry of Science and Higher Education of the Russian Federation (Agreement No. 075-15-2021-606).

**Institutional Review Board Statement:** Not applicable.

**Informed Consent Statement:** Not applicable.

**Data Availability Statement:** The data presented in this study are available on request from the corresponding authors.

**Acknowledgments:** The authors acknowledge K. A. Laptintskiy and M. V. Gulyaev for the assistance with the FTIR measurements and helpful discussions, respectively. The TEM analysis was performed using the equipment of the Center of Shared Research Facilities (MIPT). D.U.M. and A.N.K. acknowledge the Phys-Bio MEPhI and A.A. Garmash for the administrative and technical support of this research and donations in materials used for experiments.

**Conflicts of Interest:** The authors declare no conflict of interest.

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
