# Peer review of "Gadolinium-Doped Carbon Nanoparticles with Red Fluorescence and Enhanced Proton Relaxivity as Bimodal Nanoprobes for Bioimaging Applications"

_applsci, doi:10.3390/app13169322_

Round 1

Reviewer 1 Report

Comments to the Author

In the manuscript, the authors reported that the Gd-doped CNPs were prepared by microwave/hydrothermal synthesis, and applied in bimodal fluorescence/MRI bioimaging. However, some comments should be reconsidered before getting a possible publication, as follows:

1. The TEM in Figure1, please reprocess these references for reference: Materials 2023, 16, 5243; ACS Nano, 2023, 17, 9, 8671-8679; ACS Applied Materials Interfaces, 2023, 15, 28, 33868-33877; Membranes 2023, 13, 693; and Carbon, 2023, 213, 118249.

2. Please provide high-quality electron image 5 in Figure 1S. Image quality can be referenced in literature: Appl. Sci. 2023, 13, 8730; Separations 2023, 10, 429 and Small, 2023, 2303156.

3. How about the novelty of the data of Gd-CNPs prepared by microwave and hydrothermal? And it is not actually applied to bimodal fluorescence/MRI bioimaging.

Need to improve

Author Response

  1. We have improved an analysis of the TEM data in Figure1. Several recommended references were also added and discussed in the revised manuscript (pages 3: 1st and 2nd paragraphs in the Section 3; new Refs.[26,27]).
  2. We have added new TEM images and EDS maps in Fig.1S as well as ne Tables 1S and 2S with the elemental content. The corresponding discussion was added in the revised manuscript (page 4).
  3. The novelty of our work lies not in the methods of synthesis, but in the comparison of the properties of GD-doped carbon nanoparticles with the same sizes but with rather different structural properties. Thus, the nanocrystalline particles obtained by microwave synthesis have a higher PL quantum yield and an increased absorption intensity in the visible region, in comparison with the nanoparticles obtained by hydrothermal method. At the same time, the hydrothermal synthesis provides mostly amorphous nanoparticles with increased MR-relaxivity due to better coordination of gadolinium atoms on their surfaces. The novelty of our work has been additionally emphasized by providing the quantitative information on the PL efficiency and MRI contracting in the abstract of the revised manuscript.

Reviewer 2 Report

This work synthesized gadolinium doped carbon nanoparticles by microwave and hydrothermal methods. They characterized the carbon nanoparticles with TEM, FTIR and other spectroscopy methods. They showed that this type of carbon nanoparticles have the potential to be used in dual modality imaging as fluorescence and MRI contrasting agents. The authors should resolve the following issues before it is recommended for publication.

1. page2, line 79: is isopropyl alcohol aiding evaporation?

2. page 2, line 91: is “AND” typo?

3. for carbon nanoparticle synthesis, explain the reason that the two specific conditions for microwave and hydrothermal treatment were selected. It seems these two conditions are very different. Typically when doing a comparison, experimental conditions were selected to be same or similar.

4. page 4, figure 1: please show the number of particles that were counted/sized in the figure caption (n=?)

5. page 4, figure 2A: “ATR-FTIR signal” should be either transmittance or absorbance. Please check.

6. page 4, table 1: there is no C-H bond in COOH group. Also, please check how reference 26 talks about FTIR assignments. And check if reference 27 talks about FTIR at all. 

7. page 5, I’m curious about line 164-165 where HTS shows higher C=C signal. From figure 1, it seems MWS has crystalline structure while HTS doesn’t. Therefore, MWS should have more aligned C=C bond than HTS?

8. page 5, line 171: please check if it is commonly accepted that carbon hydrate refers to =C-H.

9. page 5, line 190: if QY is reported here, please show in the method section how QY was measured/estimated.

10. page 5, line 200: It’s a bit misleading to compare QY of the green emissive MWS CNP to red CDs in the literature, since it is challenging to make red CDs with high quantum yields.

11. page 6, line 210: the upconversion feature needs to be carefully discussed. The low intensity edge on the emission spectra could hardly be used in any practical applications.

12. page 6, line 236: please show the wavelength of UV excitation in the figure caption.

13. page 7, figure 4: why is the concentration range for the 2 CDs very different? Please justify if you extrapolate the linear trend line for MWS CDs.

14. in SI, figure 1: It seems the bright spots on each elemental mapping do not match the others. What does this mean? Some particles have higher C, while others have higher O or Gd?

English is fair. I have no trouble reading it, but there is always room for improvement in writing.

Author Response

  1. Indeed, the isopropyl alcohol evaporation was used to obtain the dried nanoparticles with small sizes and without byproducts. The corresponding explanation of the experimental details was added in Section 2 of the revised manuscript.
  2. We have added more accurate description of the scales manufacturing company, i.e. A&D Weighing, USA, in Section 2 of the revised manuscript.
  3. Two specific conditions for microwave and hydrothermal treatment were selected to obtain carbon nanoparticles with similar size distributions but rathe different structural properties and composition. Following to the Reviewer’s criticism we have remove “comparative” from the abstract but it was additionally emphasized the differences in the structure and composition of nanoparticles, which determines their bioimaging properties.
  4. The numbers of counted particles (N=188-189) were added in the figure caption (Fig.1)
  5. It has been corrected in page 4, figure 2A: “ATR-FTIR Absorbance”.
  6. We have corrected the assignment of C-H bond and checked the corresponding references. 
  7. We thank the Reviewer for this interesting remark. We think that the absorption by C=C bonds in HTS amorphous carbon nanoparticles cannot be unambiguously avoided because the local hexagonal structure can be preserved. Probably, a special XRD and electron diffraction experiments are required to clarify the local structure carbon mesh in amorphous nanoparticles.

8. We thank the Reviewer for this very useful remark and reference, which have been fully considered in the revised manuscript (lines 193-197, new Ref.[28])

  1. The QY measurement procedure has been added to the method section (lines 126-128).
  2. The mentioned comparison has been removed.
  3. We agree with this remark and corrected the text to ove possible practical application of the observed upconversion PL. New sentence about required further analysis of this phenomenon has been added to the revised manuscript (lines 240-242).
  4. The wavelength of UV excitation is shown in the figure caption.
  5. We have added an explanation of the different concentration ranges for MWS and HTS CDs in the caption of Fig.4.
  6. We especially acknowledge the Reviewer for this very useful remark. To follow the Reviewer’s criticism, we have additionally performed the TEM-EDX analysis, that allowed us to improve the evaluation of atomic content in the prepared samples. New Fig.1S, 2S and Table 1S with the results of these studies were added to SI. The corresponding discussion was added in the revised manuscript (lines 158-168).